# Narrative Review of Synergistics Effects of Combining Immunotherapy and Stereotactic Radiation Therapy

**DOI:** 10.3390/biomedicines10061414

**Published:** 2022-06-15

**Authors:** François Lucia, Margaux Geier, Ulrike Schick, Vincent Bourbonne

**Affiliations:** 1Radiation Oncology Department, University Hospital, 29200 Brest, France; ulrike.schick@chu-brest.fr (U.S.); vincent.bourbonne@chu-brest.fr (V.B.); 2LaTIM, INSERM, UMR 1101, University of Brest, 29200 Brest, France; 3Medical Oncology Department, University Hospital, 29200 Brest, France; margaux.geier@chu-brest.fr

**Keywords:** immunotherapy, stereotactic radiotherapy, immune-modulation, abscopal effect

## Abstract

Stereotactic radiotherapy (SRT) has become an attractive treatment modality in full bloom in recent years by presenting itself as a safe, noninvasive alternative to surgery to control primary or secondary malignancies. Although the focus has been on local tumor control as the therapeutic goal of stereotactic radiotherapy, rare but intriguing observations of abscopal (or out-of-field) effects have highlighted the exciting possibility of activating antitumor immunity using high-dose radiation. Furthermore, immunotherapy has revolutionized the treatment of several types of cancers in recent years. However, resistance to immunotherapy often develops. These observations have led researchers to combine immunotherapy with SRT in an attempt to improve outcomes. The benefits of this combination would come from the stimulation and suppression of various immune pathways. Thus, in this review, we will first discuss the immunomodulation induced by SRT with the promising results of preclinical studies on the changes in the immune balance observed after SRT. Then, we will discuss the opportunities and risks of the combination of SRT and immunotherapy with the preclinical and clinical data available in the literature. Furthermore, we will see that many perspectives are conceivable to potentiate the synergistic effects of this combination with the need for prospective studies to confirm the encouraging data.

## 1. Introduction

Cancer is a disease caused by uncontrolled cell proliferation, resulting in uncoordinated and unfaithful DNA replication, genomic instability, and DNA double-strand breaks (DSBs). For decades, the principle of using DNA damage to kill tumor cells has been applied. However, there is still the problem of how to kill the tumor but not the normal cells. In addition, cancer cells often have defects in the DNA repair mechanism, which leads to genomic instability and, thus, promotes tumorigenesis. However, these defects in DNA repair can be exploited by inhibitors that target other DNA repair pathways frequently used by cancer cells.

Indeed, to continue their proliferation, tumors will recruit stromal elements from neighboring tissues or the bloodstream, such as fibroblasts, immune cells, vascular and lymphatic endothelial cells, pericytes, and adipocytes [1]. This immunosuppressive tumor microenvironment allows cancer cells to escape immune surveillance.

Radiation therapy (RT) is a major treatment for many oncologic diseases. Although it is a local treatment, it has improved the survival and quality of life of cancer patients, even in advanced stages. On the other hand, with the emergence of immune checkpoint blockers, immunotherapy has recently become a potentially curative systemic therapy for several cancers. The interaction between these two treatment modalities with curative potential remains to be fully established. For example, some studies have shown possible immunosuppressive effects of radiotherapy [2,3,4], while others are in favor of an enhancement of anti-tumor immunity [5]. New evidence suggests that high-dose fractionated radiation therapy—known as stereotactic radiation therapy (SRT), including radiosurgery (SRS) for intracranial lesions and stereotactic body radiation therapy (SBRT) for extracranial lesions—may induce a strong antitumor immune effect [6]. Whether this effect is stronger than when using conventional normofractionated radiotherapy is not yet established, but recent data are of interest and warrant further study. In this article, we review the available data on radiotherapy-induced immune stimulation, focusing on existing data regarding local and systemic immune responses after SRT, and its combination with immunotherapy.

## 2. Stereotactic Radiation Therapy (SRT)

### 2.1. Definition

SRT relies on high precision tumor targeting and better protection of surrounding normal tissues. Thus, SRT requires a high accuracy at each step of the entire workflow, including disease staging; multidisciplinary discussion of the indications for SRT; tumor site adjusted imaging with appropriate spatial and temporal resolution for target and organ at risk (OAR) definition; highly conformal treatment; image-guided patient setup; active or passive intrafraction motion management and follow-up (preferably at the treating institution). Radiation doses are delivered in few fractions (1 to 10 fractions) with high dose per fraction (≥6 to 8 Gy). Historically, as the SRT technique used fixed circular collimators with dedicated delivery system, the dose prescription has been using the so-called “coverage isodoses” to represent some form of minimum target dose. Now, SRT can be adequately performed with either traditional linear accelerators equipped with suitable image-guidance technology, accelerators specifically adapted for SRT. Additionally, in intensity modulated radiotherapy (IMRT) for a target volume, the median absorbed dose is close to the mean absorbed dose (and should be as close as possible to the near minimum dose D98% so as to achieve dose homogeneity). However, in SRT, there is generally much less normal tissue within the planning target volume (PTV) and there may be critical normal tissues very close to the target volume. Hence, the treatment of these spatially limited volumes with multiple small photon beams may imply dose heterogeneity to maintain optimal conformity and acceptably steep dose gradients. However, there is no consensus and a high variability in dose prescription between teams and center exists [7]. Its development has opened up a range of treatment possibilities, which have already been integrated into the management of isolated targets in the brain (SRT) and extra-cranial organs (SBRT), such as the lung, spine, liver, kidney, or prostate [8,9,10,11,12,13].

The main indications of SRT are early stage tumors, which have not yet metastasized, and oligometastatic cancers with controlled primary lesions [9]. Fractionation determines a high dose-dependent tumor control rate, generally with reasonable toxicity rates [14]. For patients with oligometastatic cancer, several studies have suggested a significant improvement in survival in those treated with SRT compared to historical controls [15,16]. Unfortunately, occult microscopic tumor cell deposits may exist at other sites in patients with macroscopic oligometastases [17]. This may explain the high rate of new distant metastases after SRT (60–80%) [9,15,16]. In this context, a surprising observation was first described by Mole in 1953: the abscopal effect (from the Latin ab scopus, far from the target). To this day, this phenomenon remains rare [18]. It is defined as the further regression of other secondary tumor lesions in out of field sites after local radiotherapy, analogous to a distant bystander effect [19]. Current major advances in immunotherapy have led many studies to reconsider this potential effect of radiotherapy, especially through hypofractionated ablative irradiation. Indeed, the combination of the two treatments (immunotherapy and SRT) is widely considered as a potentially important therapeutic opportunity. However, it is essential to improve our understanding of this phenomenon to increase the chances of success of the combination of SRT and new immunotherapies.

### 2.2. Immuno-Modulation by SRT

SRT induces numerous biological responses, both at the level of the tumor itself, with notable DNA strand breaks of tumor cells that can lead to their death, but also at the level of the tumor microenvironment (TME) with the activation of several signaling pathways leading to a pro-inflammatory TME and also damage to the stroma and endothelium [20,21].

To understand the anti-tumor immune response observed after SRT, we will describe the three main mechanisms that will lead to this phenomenon (Table 1). The first is the induction of immunogenic cell death (ICD) and the release of tumor-associated antigens. The second is the alteration of the immunophenotype of the target cells. Finally, the last one is the modulation of the immune composition of the TME. Before addressing these different points, it should be remembered that several potential biases may limit the interpretation of the observed results, especially concerning preclinical studies. In particular, the radiation doses and the number of fractions are very variable and we will see that they could have a significant impact on the observed results, whether it is the local control but also the induction of a possible abscopal effect [22].

### 2.3. Activation of Dendritic Cells by Induction of Immunogenic Cell Death

Radiation-induced ICD is essential for the initiation of innate and adaptive immune responses in TME [23,24]. In ICD, distinct molecular events will occur in dying cells that are required. Initially, we observe the translocation of calreticulin from the endoplasmic reticulum to the cell surface. Then, there is a passive extracellular release of adenosine triphosphate and proteins leading to an important inflammatory response. These phenomena will allow the activation of dendritic cells (DC) [24]. These different molecules may have two roles, that of recruiting DCs and antigen-presenting cells (APCs) to the TME and that of damage-associated molecular patterns (DAMP). Note that the release of DAMPs also allows maturation of DCs leading to an APC phenotype [2]. Together, these biological responses following SRT induce a tumor-specific adaptive immune response through tumor-associated antigen (TAA) uptake and subsequent cross-presentation to CD8+ cytotoxic T cells (CTL) [25]. However, it appears that these radiation-induced events are dose-dependent [22].

Two important mediators of the immune response to radiation are the STING pathway and type 1 interferon (IFN) signaling. In a B16 melanoma model with large doses of 15 to 25 Gy in a single fraction, the University of Chicago group demonstrated that the therapeutic response to radiation therapy is dependent on induction of type 1 IFN signaling by autocrine production of IFN-β [26,27].

Enhanced cross-priming of DCs by type 1 IFNs was found, resulting in stimulation of antigen-specific adaptive T cell responses; a disappearance of these effects was shown in type 1 IFN-deficient mice, but exogenous intratumoral administration of IFN-β obviated the need for radiation-induced type 1 IFN signaling. This team continued these investigations in the same area and demonstrated a radiation-induced link that connects innate and adaptive immunity via STING, a signaling molecule involved in the innate response to cytosolic nucleic acid ligands. Deng et al. [28,29] demonstrated that the STING pathway could be activated by extruded tumor-derived DNA in response to radiation-induced ICD after penetrating the cytosol of dendritic cells. This will allow activation of IFN-β transcription, which is required for DC activation, ultimately leading to cross-presentation of TAAs and priming of tumor antigen-specific CTLs [28,29].

Other studies have also demonstrated the importance of radiation-induced type 1 IFN signaling in DC priming and effector CTL recruitment in the TME [30,31,32]. More recently, research by the Vanpouille-Box team has provided a better understanding of radiation-induced immunogenicity. Indeed, they noticed that Trex1, a DNA exonuclease, is induced in a dose-dependent manner and that it degrades cytosolic double-stranded DNA. This removes the essential substrate for activation of the STING pathway and subsequent IFN-β-mediated recruitment and activation of Batf3-dependent DCs. Thus, the potentially critical mechanism linking radiation dose and fractionation with antitumor immunity has been understood and has considerable implications for translation into the clinic and synergy with immunotherapy [33,34].

### 2.4. Upregulation of CD8+ T Cells by Enhancement of Tumor-Associated Antigen Presentation

Modulation of the expression of molecules and cell surface receptors are biological responses induced by irradiation that would increase immunogenicity and antigenicity. A study evaluated the expression of Fas, MHC class I, ICAM-1, CEA, or mucin-1 72 h after exposure to different dose levels (0, 10, or 20 Gy) of cell lines from several histological types (colon, lung, prostate) [35].

In 91% of cases, the authors observed an increase in the expression of these surface molecules. Moreover, this increase was more important with the dose of irradiation. Another interesting observation was the greater sensitivity of CEA-expressing colon cancer cells to killing by radiation-induced CEA-specific CTLs compared to the same group of non-irradiated cells. Thus, this modulation induced by high-dose irradiation could increase the immunogenicity and sensitivity of tumor cells to CTL effector functions.

One mechanism that could explain the upregulation of immunogenicity and antitumor immunity concerns the major histocompatibility complex (MHC) class I. One of these main actions is the presentation of intracellular peptides (or exogenous antigens via cross-presentation) to CD8+ T cells. SRT could induce an increase in antigen presentation on tumor cells via an increase in MHC class I expression. This biological response would be a way to avoid tumor immune escape. Indeed, escape from the immune system has been observed after a decrease in the action of MHC class I [36]. Several authors have reported an increase in tumor response to immunotherapy due to radio-induced regulation of MHC 1 [37]. They also observed the dose-dependent relationship in upregulating MHC class I expression [38].

Following irradiation, an increase in the pool of intracellular peptides is observed, allowing not only the assembly of MHC class I but also the presentation of antigens. Moreover, the increase in the intracellular peptide pool occurs in several distinct phases. A first phase in which radiation therapy induces degradation of damaged proteins and a subsequent second phase in which increased protein translation is observed via activation of the mammalian target pathway of rapamycin [37,39]. Specifically, ionizing radiation has the ability to induce modulation of the peptide repertoire. The majority of peptides loaded on MHC class I were similar between irradiated and non-irradiated tumor cells, but, importantly, radiation therapy is able to induce the expression of new proteins, thereby generating unique antigenic MHC class I peptides resulting in increased polyclonal antigen-specific CTL responses [37].

Ionizing radiation has also been shown to increase the sensitivity of tumor cells to immune-mediated cytotoxicity via the Fas/Fas ligand (FasL) pathway. This pathway plays a major role in the regulation of apoptotic cell death and in the immune surveillance used by NK (natural killer) and T cells [40]. Fas belongs to the TNF receptor superfamily and is a well-known death receptor that can induce a proapoptotic cascade when ligated with FasL. When looking at the active antitumor immune response, Fas-expressing tumor cells can be targeted and eliminated by CTLs and NK cells [41]. On the other hand, tumor cells can escape this mechanism by down-regulating Fas or FasL expression, thereby promoting immune tolerance [42,43]. Chakraborty et al. showed an increase in Fas and ICAM-1 expression after tumor irradiation on the MC38 murine colon carcinoma cell line [44]. They also demonstrated that this regulation was dose-dependent.

In addition, an increase in antigenic CTL cytotoxicity of MC38 tumor cells expressing CEA, which was limited by FasL blockade, was found after irradiation. This finding emphasizes the major role of ionizing radiation in Fas-mediated tumor cell death [44]. Several studies have demonstrated an upregulation of Fas after irradiation of 8 or 16 Gy administered as a single dose or fractionated over 4 or 8 days, respectively [25,45]. It seems, also, that an upregulation of NKG2D ligands on tumor cells is found after exposure to ionizing radiation. This makes tumor cells more sensitive to NK cell-mediated cytotoxicity [46,47].

Because of the rapid and significant entry into clinical routine of immune checkpoint blockade and the potential synergy in combination with SRT, it seems essential to understand the phenotypic and functional impact of SRT on the expression of immune checkpoint molecules. In this context, the radiation-induced modulation of immune checkpoint molecules after single-dose irradiation ranging from 0 to 15 Gy on different human prostate carcinoma lines and on normal prostate epithelial cells has been studied [48]. There was an increase in costimulatory immune checkpoint molecules and a decrease in the surface expression of co-inhibitory immune checkpoint molecules. However, there was considerable heterogeneity between the different cell lines. In addition, a radio-induced activation of CTL effector function mediated by IFN-ɣ was observed to complement the changes in expression of immune checkpoints [48].

### 2.5. Immunomodulation of the Tumor Microenvironment

Alteration of cytokine signaling and the presence of immunosuppressive cells contribute to the maintenance of a profoundly immunosuppressive TME, but with considerable intra-tumor heterogeneity [49]. SRT, like all therapeutic ionizing radiation directed against tumors, can help transform the TME into a more immunostimulatory environment. However, ionizing radiation can have the opposite effect and lead to immunosuppressive and protumorigenic effects. It is possible to simplify the modulation of the tumor immune microenvironment induced by ionizing radiation into three major steps. First, there is local production of chemokines, cytokines, and other soluble factors, then there are alterations in the tumor-associated stroma and endothelium, and, finally, there is trafficking or modulation of immune cell subsets in the TME [1].

### 2.6. Abscopal Effect

The abscopal effect is a very controversial hope in the field of radiation therapy. It is defined as a suppressor of unirradiated tumors or metastases. So is it possible to irradiate a tumor and create antitumor effects outside the irradiated field? The main hypothesis concerning the mechanism of this phenomenon is immune-mediated. Radiotherapy would reveal tumor-specific antigens which would then be recognized and captured by dendritic cells leading to activation of T cells in the neighboring lymph nodes [50]. More and more research is being performed on the abscopal effect since it was found that checkpoint inhibitors have a much higher immunomodulatory activity than activating cytokines (e.g., IL-2) or infusions of activated immune cells.

## 3. Synergistics Effects of Combining Immunotherapy and Stereotactic Radiation Therapy

SRT is a radiotherapy technique leading to very good results in terms of local control, however, many distant relapses are observed after exclusive SRT treatment, probably due to the maintenance of an immunosuppressive environment despite this SRT [51]. Thus, it seems interesting to combine inhibitors of these escape pathways to promote the development of an anti-tumor TME.

Especially since ITs, compared to other systemic treatments, are treatments with few side effects and effective on tumor cells regardless of their division rate [52]. Moreover, in patients who respond to immunotherapy, long-term responses are often observed. However, the response rate to immunotherapy alone remains too low due to low antigenic exposure of tumor cells and the presence of tumor-induced immunosuppressive TME [52]. Indeed, tumors are often classified into two categories based on the distribution of immune cells in the TME and their response to ICI. These are the hot tumors, also called immune-inflamed tumors, and the cold tumors including immune-excluded and immune-desert tumors. The first group tends to respond better to ICIs through a higher T-cell infiltration into the TME, higher tumor mutational burden, and increased IFN-γ signaling and PD-L1 expression [53].

These different observations led to the idea of combining SRT and IT for their potential synergistic action, with in particular an increase in antigenic expression of tumor cells induced by SRT allowing resensitization of tumors to IT.

Currently, two immune system pathways have been studied in clinical trials, CTLA-4 and PD-1.

### 3.1. Rationale and Preclinical Data

As explained above, SRT can induce ICD, TAA release, and increased antigen presentation. DAMPs will allow activation and maturation of DCs that then migrate to the tumor-associated draining lymph nodes [23,54]. In the lymph node, two signals allow efficient activation of T cells. The first is through binding between antigens presented on the MHC class I and the antigen-specific T-cell receptor on a CD8+ T-cell. The second signal involves CD80 or CD86 on the APC and CD28 on the CD8+ T cell. However, cytotoxic T cell-associated antigen 4 (CTLA-4) is an important negative regulator of T cell activation. Indeed, it will compete directly with CD28 for binding to CD80 and CD86. In addition, CTLA-4 has a higher affinity than CD28 for CD80 and CD86, allowing it to curb excessive or inappropriate T cell activation, which can lead to autoimmunity and other deleterious effects [55]. Blocking CTLA-4 is possible with a monoclonal antibody to shift the balance in favor of T cell activation [52]. This anti-CTLA-4 monoclonal antibody was the first immune checkpoint inhibitor (ICI) in its class to receive U.S. Food and Drug Administration approval for metastatic melanoma in 2011 [56]. In addition, preclinical reports have shown that anti-CTLA-4 therapies act synergistically with radiation therapy [57,58]. The main hypothesis is the action of anti-CTLA-4 and SRT on immunosuppressive regulatory T cells (Tregs) [59,60]. Indeed, anti-CTLA-4 will potentiate pro-immunogenic effects (i.e., increase CTL activation) and limit immunosuppressive elements (i.e., depletion of intratumoral Tregs) in TME.

In preclinical studies, the administration of an anti-CTLA-4 antibody subsequently to SBRT (two fractions of 12 Gy) was shown to significantly improve survival in breast cancer-bearing mice compared to the checkpoint inhibition alone—by preventing the formation of lung metastases [61].

Usually, once activated, CTLs migrate into the TME to perform their anti-tumor action. However, mechanisms of resistance to their cytotoxic action have been observed in some tumors. One of the mechanisms of immune escape is through the PD-1/PD-L1 (programmed cell death protein 1 or programmed cell death-ligand 1) axis. This pathway has therefore been studied in order to swing the TME towards greater antitumor immunity [62]. PD-1 is an immunoinhibitory receptor whose expression is found on many cells of the immune system (T cells, NK cells, B cells and DCs). It should be noted that T cells express PD-1 only after engagement of the T cell receptor (TCR), thus, naive and resting T cells do not express PD-1. Finally, a subset of tumor antigen-specific tumor-infiltrating lymphocytes (TILs) was found to be deficient in tumors. This subset of TILs expressed PD-1 [63]. PD-L1 is the major ligand of PD-1. At the tumor level, its expression is found not only in tumor cells but also in TILs and other myeloid-derived cells with an increase in this expression in many tumor types. This is a mechanism of immune escape. Indeed, it will allow PD-1 to be engaged leading to a series of immunosuppressive phenomena, including T cell death or the production of immunosuppressive IL-10 [62,63]. However, with the emergence of therapies targeting this pathway, PD-L1 overexpression is reported to be a predictive biomarker of response to therapy and a good prognostic factor in certain tumor subtypes [64]. Studies have also shown a very interesting response to irradiation in this context. Indeed, RT seems to increase the expression of PD-L1 which would increase tumor sensitivity to IT [5]. However, since PD-L1 expression is also found in the myeloid cells that are essential for a good response to treatment, it is possible that irradiation induces immunosuppression [63]. However, preclinical data on the combination of SRT and PD-1/PD-L1 axis blockade are reassuring, and it seems that SRT tips the balance towards anti-tumor TME [65,66].

A hybrid (preclinical and clinical) study reported interesting results on the combination of SRT with dual immune checkpoint blockade by PD-L1 and CTLA-4. Indeed, this combination overcame immune tolerance by non-redundant mechanisms [67]. Other co-inhibitory immune checkpoint pathways (i.e., TIM-3, VISTA, LAG-3, TIGIT) but also T cell costimulatory pathways (i.e., OX40, 4-1BB) are under evaluation [68,69].

The extent of the immune infiltrate and the balance between immunostimulatory and immunosuppressive subsets may impact the response to therapy [70]. CTL-mediated antitumor effects may be reduced by suppressive Tregs, myeloid-derived suppressor cells (MDSCs), and tumor-associated macrophages. In addition, anti-tumor immune responses may be further reduced by local production of immunosuppressive cytokines, including IL-10 and TGF-β, by tumor cells or associated stroma. Irradiation of the tumor will induce the production of proinflammatory cytokines (i.e., IL-1β and TNF-α) and increase the infiltration of immunostimulatory T cells via chemokine signaling [71]. The expression of CXCL9 and CXCL10, which serves as a chemotactic gradient to recruit T cells expressing the corresponding receptor CXCR3 into the TME will be stimulated by IFN-β signaling [72]. In addition, a mouse model of 4T1 breast cancer observed that CXCL16, that recruits and binds CD8+ T cells expressing CXCR6, is specifically released in response to tumor irradiation [73]. In addition, the therapeutic model investigated the association of ionizing radiation with anti-CTLA-4 blockade. They observed that in CXCR6-deficient mice, the ratio and density of CD8+ TILs were significantly lower [73].

### 3.2. Clinical Results of Combination of IT and SRS for Brain Metastases

The definition of ICI-SRS concomitance varies widely among studies. The vast majority consider concomitant to be the administration of SRS within 4 weeks before or after the onset of ICI [74,75,76,77,78,79,80,81]. However, some authors take <2 weeks [82,83] and others up to >2 months [84,85].

Furthermore, in studies that compared exclusive SRS and SRS + ICI, the definition of exclusive SRS was also variable between studies. Some considered only patients who had not received ICI [86,87]. However, other authors included in the exclusive SRS subgroup patients who had received a last ICI at least 3 months [74] or 6 months before [88].

In studies comparing exclusive SRS vs. ICI-SRS, some groups have reported improved local control (LC) with the combined treatment [79,88]. For example, Acharya et al. found that combination SRS + ICI (including anti-PD-1 and anti-CTLA-4) was associated with a significant decrease in local failure (LF) compared with SRS alone [hazard ratio (HR) 0.37; 95% CI: 0.14–0.95; *p* = 0.04] in melanoma brain metastases [51]. However, other studies found no difference [75,82,85,86] (Table 2).

Results were also reported according to the systemic treatment administered (anti-PD-1 vs. anti-CTLA4). In patients treated for melanoma, the rate of LC appears to be higher when using anti-PD-1, with rates of 80–96% [79,83,87,89] compared to anti-CTLA4, with rates of 16.5–100% [74,75,85]. Minniti G. et al. even showed a statistically significant increase in LC when anti-PD-1 was used compared to anti-CTLA4 (85% versus 70%, respectively) [82].

When looking at the sequence of treatment (concurrent or sequential), the results in terms of LC were highly variable. Some studies found no difference [76,81]. However, one study found a trend in favor of concomitant treatment over sequential treatment (88% vs. 79%, respectively; *p* = 0.08) [83] and one study reported a significantly higher 1-year LC in favor of concomitant (54.4% vs. 16.5%; *p* < 0.05) [75].

For overall survival, some studies found an improvement in OS with ICI-SRS combination compared to SRS alone [76,87], or with concomitant treatment compared to sequential treatment or SRS alone [74,75,82,85].

For progression-free survival (PFS), the combination of ICI-SRS versus SRS alone may also provide benefit [76,77,84].

Concerning toxicity, studies have reported G3+ toxicity ranging from 5% to 24% and no G5 toxicity but the results are mixed to answer the question if the combination of SRS + ICI increases the risk of toxicity and especially radionecrosis.

For example, Chen et al. compared the results of 260 patients treated with either concomitant SRS + ICI (anti-CTLA-4 or anti-PD-1) or non-concomitant SRS + ICI and found no difference in toxicity [83]. However, Martin et al. studied the results in 480 patients treated with SRS with or without ICI. They noted an increased risk of symptomatic radionecrosis in the SRS + ICI group (HR 2.56, 95% CI: 1.35–4.86, *p* = 0.004).

Only one study reported a case of abscopal response in SRS [89].

### 3.3. The Combination of ICI and SBRT for Extracerebral Lesions

Fractionation patterns vary from study to study, but multiple fractionation (3 to 10 fractions) is the most common treatment modality [90,91,92,93,94].

The LC rate after SBRT + ICI treatment varies from 40% to 94%. However, if only prospective clinical trials are considered, it increases to 75–91% [90,91,92,93,94].

In the PEMBRO-RT study, patients included had a recurrent metastatic non-small cell lung cancer (NSCLC). The main objective of this study was to evaluate the impact of the combination of pembrolizumab with 24 Gy of SBRT in three fractions on a single metastatic site compared to treatment with IT alone.

The authors reported very encouraging results in favor of the SBRT + IT combination with a doubling of the response rate at 12 weeks, an improvement in progression-free survival (6.6 months vs. 1.9 months) and overall survival (15.9 months vs. 7.6 months). The subgroup that appeared to benefit most from the addition of SBRT was patients with PD-L1 negative expression. This result supports the hypothesis that SBRT may induce PD-L1 overexpression and sensitize tumors unresponsive to IT alone by the absence of PD-L1 expression [94].

Reported abscopal response rates range from 10% to 45%. Sundahl N. et al. found 45% responses in non-irradiated lesions and 15% complete response [91].

A major limitation in the interpretation of the data in the literature, particularly concerning local control or induction of an abscopal effect, is that patients were treated at many different sites (lung, liver, bone). Thus, it is difficult to analyze the impact of the location on the results for the majority of the studies. However, some authors have reported results on this issue. A first study found greater activation of the immune system in the case of liver irradiation than in the case of lung irradiation. The authors hypothesized an earlier increase in CD8+ T lymphocytes as well as a greater overexpression of PD-1 on these cells [92]. A second team reported different results with no role of the irradiation site. However, they reported the predictive value of some genes associated with IFN expression for the abscopal effect [93].

Only one study investigated the difference in outcome according to ICI [95]. Anti-PD-1 combined with SBRT appeared to improve abscopal response (37% vs. 24%), overall survival (NA vs. 10.7 months), and disease-free survival (NA vs. 6.4 months) significantly compared with the combination of anti CTLA-4 and SBRT. No studies examined the impact of the timing of SBRT versus ICI administration.

Given the relatively novel concept of combining ICI with SBRT in patients with metastatic disease, the safety of combination therapy is of significant concern, especially given the uncertain clinical benefit of this regimen.

A Phase 1 study evaluated the combination of SBRT + pembrolizumab in patients treated for various metastatic solid tumors. Of the 79 patients included, 68 were treated with both SBRT and IT.

Patients initially received SBRT at one to four metastatic sites and then received a pembrolizumab injection every three weeks until disease progression (clinical or radiographic), dose-limiting toxicity, study withdrawal or death was reported. Dose-limiting toxicity was defined as a CTCAE grade 3 or higher adverse event. Six cases were noted. The conclusion of the study was that SBRT followed by pembrolizumab had an acceptable toxicity profile [93].

In prospective clinical trials, the rate of grade 3 or higher toxicity ranged from 0% to 34% [90,91,92].

Thus, the results in the few prospective trials show high local control and an encouraging abscopal response rate (>10%) with acceptable toxicity profile. However, larger prospective clinical trials are needed to answer many questions.

### 3.4. Ongoing Studies

As mentioned above, few Phase 2 and Phase 3 trial results are available at this time. However, many trials are either enrolling or under analysis with data soon to be available. For example, a trial evaluating SBRT + pembrolizumab in patients with melanoma or NSCLC previously refractory to ICI (NCT03693014) [96]. In addition, a trial is currently underway in patients with metastatic NSCLC, melanoma, renal cell carcinoma, or head and neck cancer who are randomized between treatment with pembrolizumab alone or pembrolizumab plus SBRT (NCT02318771) [97]. Similarly, one study includes patients with metastatic NSCLC who have already received first-line therapy. It randomized patients between pembrolizumab alone and pembrolizumab + SBRT in 3 to 10 fractions for a total dose of 30 to 60 Gy (NCT03867175) [98].

However, the combination of SBRT and immunotherapy is also of interest to patients with non-metastatic cancer, including unresectable stage I or II NSCLC. A study is currently randomizing patients with recurrent stage I or II NSCLC between SBRT alone and SBRT + nivolumab every 2 weeks for 3 months, barring dose-limiting toxicity. The endpoint was event-free survival (including local recurrence, regional recurrence, distant metastasis, secondary malignancy, and death (NCT03110978) [99].

There is also PACIFIC-004 and SWOG/NRG S1914, two ongoing multicenter phase 3 trials. They focus on non-surgical NSCLC. They compare SBRT + placebo to SBRT + ICI, durvalumab every 4 weeks for 24 months or until one of the stopping criteria is met (NCT03833154) [100] or atezolizumab every 3 weeks for eight cycles (NCT04214262) [101].

In these studies, the biologically effective doses (BEDs) are greater than 100 Gy and will provide safety data on the use of ablative doses of radiation therapy in combination with ICI.

Several clinical trials which will provide more data regarding combined ICI-SRS therapy are currently underway, such as the MIGRAINE trial (NCT04427228) [102] and STICk-IM-NSCLC (NCT04650490) [103]. The first compares multifractionated SRS (27 Gy in three fractions) with monofractionated SRS (18 Gy for lesions larger than 2 cm and 20 Gy for lesions smaller than 2 cm). All patients received concomitant ICI. The primary objective is the rate of radionecrosis to determine whether multifractionated SRS is safer. The second evaluates the timing of SRS versus ICI in NSCLC patients with brain metastases. Patients are randomized between SRS followed by ICI within 14 days and ICI with SRS only.

## 4. Perspectives

### 4.1. Manipulation of the Tumor Microenvironment with New Immunotherapies

In addition to the need for prospective randomized trials combining SRT and ICI, research is also needed to improve the therapeutic index of SBRT and ICI combinations.

A first avenue seeks to manipulate the TME to enhance the immunogenic side of SRT and reduce its immunosuppressive action (Table 3).

One target of TME is tumor-associated macrophage (TAMs). Studies are looking to steer TAMs toward an M1-type antitumor phenotype and away from an M2-type pro-tumor growth phenotype. this would enhance the effect of ICI and SRT.

For example, results from studies combining granulocyte colony-stimulating factor (GM-CSF) with RT are encouraging. A trial including patients with metastatic solid tumors received the combination of GM-CSF with radiation therapy (35 Gy in 10 daily fractions). Abscopal responses were observed in untreated metastases in 11 (26.8%, 95% CI: 14.2–42.9%) of 41 included patients [104].

Furthermore, on the one hand, it has been observed that high-dose radiotherapy has an immunosuppressive action by inducing an M2-like phenotype [105,106] and, on the other hand, SRT has a pro-immunogenic action with a decrease in CD8+ T-cell response and IFN production, allowing tumor overexpression of PD-L1 [5]. Thus, manipulation of TAM polarization could make the SRT + ICI combination more effective by inhibiting the immunosuppressive action of SRT. One study focuses on combining SBRT and nivolumab with cabiralizumab (NCT03431948) [107], a colony-stimulating factor 1 receptor (CSF-1R) antagonist monoclonal antibody, which has been shown to steer TAM populations toward an M1-like phenotype [108]. The combination of cabiralizumab with nivolumab has already been studied in patients with pancreatic cancer [109]. Thus, future studies will continue to focus on directing TAMs to an antitumor phenotype to improve the therapeutic index of the SBRT + ICI combination.

Another target that may impact the balance of TME towards an antitumor phenotype is 4-1BB, a transmembrane glycoprotein presents on activated effector T cells. It increases the activity and survival of CD8+ T cells and inhibits Tregs, in response to 4-1BB ligand (4-1BBL, CD137) on APCs. Preclinical studies have shown the potential of targeting 4-1BB. Indeed, the combination of 4-1BB agonist monoclonal antibodies with radiotherapy resulted in a significant increase in response rates in mouse models of breast and lung carcinoma [110]. The 4-1BB monoclonal antibodies have also been studied in combination with PD-1 blockade and SBRT in murine models of melanoma with encouraging response rates [111]. A preclinical study in a mouse model of glioma also evaluated the combination of SRT (single dose of 10 Gy) with CTLA-4 blockade and the co-stimulatory molecule 4-1BB (CD137). This triple therapy improved tumor-free survival by more than 50% compared with radiotherapy alone. A significant increase in CD4+ and CD8+ T cells in TME has also been shown [112]. Clinical data studying the combination of a 4-1BBL agonist with radiotherapy [113] and PD-1 blockade [114] have reported sometimes impressive responses. Thus, studies combining ICI + SBRT with 4-1BB targeting of activated effector T cells seem promising. Currently, a study is underway of the addition of a 4-1BBL agonist, urelumab, to the SBRT + nivolumab combination in patients with advanced solid tumors (NCT03431948) [107].

Other targets in the tumor microenvironment that would enhance antitumor activity are currently under investigation. For example, TGF-β that induces an immune exclusion phenotype and resulting resistance to PD-L1 [115]. In this setting, studies have shown that adding TGF-β inhibition to PD-1 inhibition via bispecific antibodies targeting both PD-1 and TGF-β [116,117] was a way to overcome PD-L1 resistance. Furthermore, TGF-β is activated by irradiation. It leads to a decrease in the cross-priming ability of DCs, an inhibition of the effector function of T cells and induces the conversion of Tregs, leading to a strong immunosuppressive potential [4]. Preclinical data have shown that TGF-β inhibition in combination with radiotherapy alone, but also with multimodal SRT + ICI treatment, provides very good results [118,119]. Studies are underway on the combination of a monoclonal antibody targeting TGF-β with SRT in patients with metastatic breast cancer [120] and early stage non-small cell lung cancer (NCT02581787) [121].

In addition, studies have looked at approaches combining PD-1/PD-L1 blockade with agents targeting effector T cells and chemokine inhibition, as well as vaccine therapy to overcome PD-L1 resistance [122].

Thus, all of these strategies with the goal of inducing a more favorable TME for immune and antitumor surveillance have shown very encouraging data to potentially improve the efficacy of the SBRT + ICI combination.

### 4.2. Improvement of SRT

Questions regarding the dose and fractionation of SRT, but also the best timing of the combination and the target of SRT remain major issues to improve the therapeutic index of ICI + SRT.

#### 4.2.1. Is There an Optimal Dose and Fractionation Timing for SRT?

The results of preclinical studies regarding optimal dose and fractionation are mixed [123].

This question has recently been revived following the publication of studies that observed abscopal responses only with a hypofractionated regimen (three fractions of 8 Gy) in combination with ICI [33,34]. The underlying mechanism found was accumulation of cytosolic double-stranded DNA and activation of the STING or type I IFN signaling pathway, whereas single fraction regimens of 20–30 Gy did not achieve remote tumor control, due to the induction of Trex1, which degrades cytosolic DNA in a dose-dependent manner. Indeed, it was observed that the higher the dose per fraction the more Trex1 was induced leading to DNA degradation. Thus, as soon as the dose per fraction reaches the threshold for Trex1 induction, this leads to downstream abrogation of IFN-b production, decreased DC recruitment or activation and failure to generate systemic antitumor immune responses. These observations may suggest that radiation doses >12 Gy per fraction decrease immunogenicity via induction of Trex1 and that hypofractionated regimens may be better alternatives for combination SRT + ICI. These results could allow for better selection of the radiotherapy regimen, but these results need to be validated in the clinic.

The C4-MOSART trial, on the other hand, is looking at the optimal dose and fractionation of SBRT for the SBRT + ICI combination. Several ICIs, including urelumab, cabiralizumab, and nivolumab, are combined with SBRT depending on the primary tumor (NCT03431948) [107].

However, the therapeutic objective is also essential in the choice of dose and fractionation. Indeed, it is essential to know whether one wishes to obtain the best locoregional control or to modify the natural history of the disease by targeting metastatic sites. Clinical findings on which dose and fractionation result in local and/or systemic immunogenic effects will guide the prescription of SBRT [124].

#### 4.2.2. Is There an Optimal Target for SRT?

Another important point is that the immunologic response to SBRT is likely to be influenced by the histology of the tumor and the immune microenvironment of the different tissues or organs [125].

Thus, the question about the most appropriate target volume of irradiation arises.

First, whether certain organs or sites of disease are more or less immunogenic is critical to determining which sites should be preferentially irradiated in oligometastatic disease.

Second, the question of whether or not to irradiate regional lymph nodes in clinically localized disease remains open. Indeed, numerous reports have identified draining lymph nodes as critical for activation and accumulation of radiation-induced CTLs and generation of adaptive immune responses [38,57].

For metastatic patients, the feasibility of multi-site SRT has been made possible due to recent innovations [126].

A first advantage of this multi-site SBRT would be to reduce the antigenic load to avoid T-cell exhaustion [127]. In addition, IT has been shown to be more effective in low-volume disease [128]. Another theoretical advantage would be to increase the volume and diversity of tumor antigens released.

Another approach to improve the response to SRT would be to combine it with an angiogenesis inhibitor to increase the sensitivity of hypoxic lesions [129].

Innovations are ongoing in RT that could help improve the efficacy of the SRT + ICI combination such as biologically guided radiotherapy [130] or reduction in irradiation volumes to healthy tissue to decrease post-treatment lymphopenia [131] which could lead to a decrease in the efficacy of ICI [132].

#### 4.2.3. Who Benefits from the SRT + ICI Combination?

A key issue for improving the therapeutic ratio of ICI + SRT would be to determine the subpopulation of patients who benefit the most.

For example, molecular biomarkers have been identified that differentiate between patients treated with ICI who will have oligometastatic progression and those who will have polymetastatic progression [133]. Thus, one could consider proposing SRT only at risk of oligoprogression.

In line with this idea, one group has proposed a molecular classification to stratify patients according to the risk of failure after metastasectomy in the setting of colorectal cancer liver metastases [134]. These results provide interesting food for thought regarding the identification of patients with curable oligometastatic disease.

## 5. Conclusions

SRT is a rapidly growing radiotherapy technique because of its many advantages over standard normofractionated radiotherapy. Mainly, a better preservation of healthy tissues thanks to its high precision, better results in terms of local control and a shorter treatment time. Thus, this modality is used in the different indications of radiotherapy, curative and palliative. As we have just seen, another potential advantage of high-dose fractional RT would be its greater ability to induce anti-tumor immunity. The first results in clinical studies are hopeful but longer follow-up and additional studies are needed.

Numerous studies are underway, in metastatic and also non metastatic patients, that will provide answers to many questions.

Finally, new strategies seem necessary to maximize the therapeutic ratio of combined-modality therapy.

## Figures and Tables

**Table 1 biomedicines-10-01414-t001:** Main mechanisms of immuno-modulation by stereotactic radiation therapy.

Steps of Action	Mechanism of Stereotactic Radiation Therapy
Activation of dendritic cells by induction of immunogenic cell death	Induction of STING pathway and type 1 interferon
Upregulation of CD8+ T cells by enhancement of tumor-associated antigen presentation	Increase the expression of surface molecules (Fas, MHC class I, ICAM-1, CEA, or mucin)
Immunomodulation of the tumor microenvironment	Induction of local production of chemokines, cytokines, and other soluble factors Alterations in the tumor-associated stroma and endothelium Trafficking or modulation of immune cell subsets in the tumor microenvironment

**Table 2 biomedicines-10-01414-t002:** Level of evidence for safety and efficacy of the combination immune checkpoint inhibitor (anti-CTLA-4 and anti-PD1/PDL-1) and stereotactic radiation therapy.

Type of Stereotactic Radiation Therapy	Safety	Efficacy
SRS	Only retrospective studies found grade 3+ toxicity ranging from 5% to 24% and no grade 5 toxicity	Only retrospective studies showing variable results on the improvement of efficacy with a trend in favor of anti-PD1/PD-L1 compared to anti-CTLA-4
SBRT	Phase 1 showed an acceptable toxicity profile with anti-PD1 In prospective clinical trials, the rate of grade 3 or higher toxicity ranged from 0 to 34%	Prospective studies showing improved outcomes with the combination with greater benefit with anti-PD1 compared to anti-CTLA-4

Abbreviations: cytotoxic T cell-associated antigen 4 = CTLA-4, programmed cell death protein 1 = PD-1, programmed cell death-ligand 1 = PD-L1, radiosurgery = SRS, stereotactic body radiation therapy = SBRT.

**Table 3 biomedicines-10-01414-t003:** Perspectives of the combination immune checkpoint inhibitor and stereotactic radiation therapy (SRT).

Aims	Targets
Manipulation of the tumor microenvironment to enhance the immunogenic side of SRT	Tumor-associated macrophage Combination with granulocyte colony-stimulating factor Combination with a colony-stimulating factor 1 receptor antagonist monoclonal antibody 4-1BB, a transmembrane glycoprotein presents on activated effector T cells Combination with a 4-1BB agonist monoclonal antibodies TGF-β Combination with a monoclonal antibody targeting TGF-β Combination with a bispecific antibodies targeting both PD-1 and TGF-β
Improvement of SRT	Optimization of dose and fractionation of SRT Determine the optimal target to obtain an immunologic response Identification of molecular biomarkers to select the subpopulation who benefit the most of the combination

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
