# Peer review of "Narrative Review of Synergistics Effects of Combining Immunotherapy and Stereotactic Radiation Therapy"

_biomedicines, 2022, doi:10.3390/biomedicines10061414_

Round 1
Reviewer 1 Report
This is an interesting overview of combining immunotherapy and radiation therapy. I have the following suggestions to improve the quality of this manuscript:
Figures and Tables should be included to visualize the most important princliples and important studies.
A list of abbreviations should be included to improve readability.
The concept of "cold tumors" and "hot tumors" should be mentioned.
The definition of SRT must be improved (high precision and hypofractionation is very general) - dose per fraction? IGRT? dose homogeneity?
Author Response
We thank the reviewer for his comments

Reviewer 2 Report
This narrative review by Lucia F. et al. describes the present status of radiotherapy plus immunotherapy for caner treatment. It presents published data from several laboratory investigations and clinical trials that have attempted combining SRT with immunotherapies such as CTL-mediated and PD-1-mediated. It is a concise summary and can serve as a quick reference if published. Being a review, I do not have many comments on the manuscript in terms of its scope. However, it needs improvements in its presentation of the topic with references placed for the reader to follow. There are also a couple of places where apparently conflicting statements seem to have been made.
Major
1. In Section 4.1 (on Manipulation of the TME), I found the following statements confounding and conflicting. The authors state in the beginning that “Studies are looking to steer TAMs toward an M1-type antitumor phenotype and away from an M2-type pro-tumor growth phenotype. this would enhance the effect of ICI and SRT”. In next paragraphs they write that “Furthermore, high-dose radiotherapy has been observed to induce an M2-like phenotype [104, 105] which would make manipulation of TAM polarization even more effective in combination with SRT”. Next they state that “One study focuses on combining SBRT and nivolumab with cabiralizumab (NCT03431948) [106], a colony-stimulating factor 1 receptor (CSF-1R) antagonist monoclonal antibody, which has been shown to steer TAM populations toward an M2-like phenotype [107]”. Can the authors please clarify this?
Minor:
1. Some statements are lacking citations hindering the readers from checking the original papers. They are: (i) In addition, a radio-induced activation of CTL effector function mediated by IFN-É£ was observed to complement the changes in expression of immune checkpoints. (ii) In addition, a mouse model of 4T1 breast cancer observed that CXCL16 is specifically released in response to tumor irradiation.
2. The following sentence is incomplete: “CXCL16 that recruits and binds CD8+ T cells expressing CXCR6.”
3. The statement “The safety of ICI + SBRT in patients with metastatic disease is a significant concern, especially as it is very new.” is redundant and can be deleted.
4. The following group of short paragraphs are very choppy.
3.2. Clinical results of combination of IT and SRS for brain metastases
The definition of ICI-SRS concomitance varies widely among studies. The vast majority consider concomitant to be the administration of SRS within 4 weeks before or after the onset of ICI [71-78].
However, some authors take <2 weeks [79, 80] and others up to >2 months [81, 82].
Furthermore, in studies that compared exclusive SRS and SRS + ICI, the definition of exclusive SRS was also variable between studies.
Some considered only patients who had not received ICI [83, 84]. However, other authors included in the exclusive SRS subgroup patients who had received a last ICI at least 3 months [71] or 6 months before [85].
They can be rewritten as shown below for smoother reading:
3.2. Clinical results of combination of IT and SRS for brain metastases
The definition of ICI-SRS concomitance varies widely among studies. The vast majority consider concomitant to be the administration of SRS within 4 weeks before or after the onset of ICI [71-78]. However, some authors take <2 weeks [79, 80] and others up to >2 months [81, 82].
Furthermore, in studies that compared exclusive SRS and SRS + ICI, the definition of exclusive SRS was also variable between studies. Some considered only patients who had not received ICI [83, 84]. However, other authors included in the exclusive SRS subgroup patients who had received a last ICI at least 3 months [71] or 6 months before [85].
Author Response
We thank the reviewer for his comments
Kind regards

Reviewer 3 Report
The present paper is a narrative review regarding the available data on radiotherapy-induced immune stimulation, focusing on existing data regarding local and systemic immune responses after stereotactic radiotherapy, and its combination with immunotherapy.
The present paper is interesting, well written and exhaustive, and the topic is novel and contemporary: I only suggest to cite the follownig relevant papers on interation between stereotactic radiotherapy and systemic treatments:
Mazzola R, Tebano U, Aiello D, Paola GD, Giaj-Levra N, Ricchetti F, Fersino S, Fiorentino A, Ruggieri R, Alongi F. Increased efficacy of stereotactic ablative radiation therapy after bevacizumab in lung oligometastases from colon cancer. Tumori. 2018 Dec;104(6):423-428. doi: 10.5301/tj.5000701.
Author Response

(The authors gave the same response as above.)

Round 2
Reviewer 1 Report
The comments have been well addressed and the quality of the manuscript improved. It is acceptable for publication.